# Complete Genomic Sequence of *Xanthomonas oryzae* pv. *oryzae* Strain, LA20, for Studying Resurgence of Rice Bacterial Blight in the Yangtze River Region, China

**DOI:** 10.3390/ijms24098132

**Published:** 2023-05-01

**Authors:** Yuxuan Hou, Yan Liang, Changdeng Yang, Zhijuan Ji, Yuxiang Zeng, Guanghao Li, Zhiguo E

**Affiliations:** State Key Laboratory of Rice Biology and Breeding, China National Rice Research Institute, Hangzhou 311400, China

**Keywords:** bacterial blight, resurgence, *Xanthomonas oryzae* pv. *oryzae*, genome

## Abstract

*Xanthomonas oryzae* pv. *oryzae* (*Xoo*) is a causative agent of rice bacterial blight (BB). In 2020–2022, BB re-emerged, and there was a break out in the Yangtze River area, China. The pandemic *Xoo* strain, LA20, was isolated and identified from cultivar Quanyou1606 and demonstrated to be the Chinese R9 *Xoo* strain, which is able to override the widely adopted *xa5*-, *Xa7*- and *xa13*-mediated resistance in rice varieties in Yangtze River. Here, we report the complete genome of LA20 by PacBio and Illumina sequencing. The assembled genome consists of one circular chromosome of 4,960,087 bp, sharing 99.65% sequence identity with the traditional representative strain, YC11 (R5), in the Yangtze River. Comparative genome analysis of LA20 and YC11 revealed the obvious variability in *Tal* genes (the uppermost virulence determinants) in numbers and sequences. Particularly, six *Tal* genes were only found in LA20, but not in YC11, among which *Tal1b* (*pthXo1*)/*Tal4* (*pthXo6*), along with the lost one, *pthXo3* (*avrXa7*), might be the major factors for LA20 to overcome *xa5*-, *Xa7*- and *xa13*-mediated resistance, thus, leading to the resurgence of BB. This complete genome of the new pandemic *Xoo* strain will provide novel insights into pathogen evolution, the traits of pathogenicity on genomic level and the epidemic disease status in China.

## 1. Introduction

The phytopathogenic *Xanthomonas oryzae* pv. *oryzae* (*Xoo*) causes rice bacterial blight (BB), one of the most devastating diseases of rice worldwide, usually resulting in 10% to 30% yield loss and even 50%, or the loss of the harvest [1]. BB epidemics have been reported in 28 Chinese provinces since it was initially observed in Jiangsu province, the Yangtze River region, China, in the 1930s [2]. By the 1990s, BB was rampant in the vast rice-growing area, especially in the Yangtze River area. Until the early 21st century, the wide utilization of BB resistance (*R*) genes, such as *xa5, Xa7 and xa13,* in rice breeding effectively controlled BB, and the disease barely occurred in rice fields in the Yangtze River region [3,4]. However, due to the co-evolution of *Xoo* and rice [5,6], *R* gene-conferred BB resistance was gradually conquered by emerging *Xoo* strains [7,8]. In 2020–2022, BB had re-emerged, and there were widespread epidemics in the Anhui and Zhejiang provinces in the Yangtze River region.

TALEs (transcription activator-like effectors), the most important virulence factors of *Xoo,* function as transcription activators and bind to the promoters of susceptibility (*S*) or *R* genes of rice to regulate their transcription, which plays crucial roles in pathogenicity and variety discrimination in *Xoo*–rice interaction [9,10]. A typical TALE consists of three parts: the N-terminal domain containing the type III secretion signal, the carboxyl-terminal nuclear localization signal (NLS) and the transcription activation domain (AD) and the central highly conserved repeat units [11]. The promoter-binding specificity of TALEs is determined by repeat units. Each repeat unit has near-perfect 33 to 35 amino acid repeats, with two variable amino acids at position 12 and 13 (termed the ‘RVD’, for repeat-variable diresidue) [12,13]. TALE-encoding genes (*Tal* genes) are presumed to be highly dynamic and varied in RVDs and have greatly contributed to the production of new toxicity and the escape of plant immunity by *Xoo* [14,15]. Tracking and identifying the variations in *Tal* genes are helpful for accurate pathotype discrimination of the emerging *Xoo* strains and rice-disease-resistance breeding.

In this study, the newly emerging *Xoo* strain, LA20, was isolated from typical diseased rice leaves collected in Anhui province in 2021. We focus on the pathotype, genome sequence and TALEs dissections to uncover the reason for the BB re-outbreak in the Yangtze River region, China.

## 2. Results

### 2.1. Pathotype Analysis of Xoo Strain LA20

BB erupted in Anhui province, the Yangtze River region, China, in 2021, and the rate of incidence was up to 85% (Figure 1a). The isolated *Xoo* strain, LA20, was confirmed to be the causative agent by morphology and molecular identification (Figure 1b,c).

Upon pathotype analysis, LA20 clearly showed the “SSSSSS” interaction with six differential rice varieties (near-isogenic lines harboring individual resistance gene) IRBB2 (*Xa2*), IRBB3 (*Xa3*), IRBB5 (*xa5*), IRBB13 (*xa13*), IRBB14 (*Xa14*) and IR24 (*Xa18*) (Table 1). LA20 was demonstrated to be race 9 (R9) according to the pathotype criteria of Chinese *Xoo* strains [16,17]. However, the traditional representative strain, YC11 (R5), from the Yangtze River region showed an “SSRRSS” interaction with different rice varieties, as it failed to overcome *xa5*- and *xa13*-mediated BB resistance [18]. In addition, it was found that IRBB7 (*Xa7*) was also susceptible to LA20 and resistant to YC11 (Figure 1d–f). The results showed that LA20 is a newly discovered hyperpathogenic pandemic strain that is different from YC11 and is able to overcome widely adopted *xa5-*, *Xa7-* and *xa13*-mediated resistance. LA20 has already been deposited in the China General Microbiological Culture Collection Center (No. 25881).

### 2.2. Complete Genome Characteristics of Xoo Strain, LA20

To provide better insight into the pathogenicity, adaptation and evolution that caused BB resurgence in the Yangtze River region, the complete genome of LA20 was sequenced. In total, 336,972 reads from PacBio sequencing were obtained, with a mean concordance of 0.89, N50 value of 12,126 bp and average read length of 10,351 bp, consisting of whole 3,488,085,182 bases. High-quality filtered reads were assembled, corrected and annotated. As a result, one unique circular chromosome of 4,960,087 bp was obtained with 63.69% GC content, and the average nucleotide identity (ANI) rates are up to 99.67% with *Xoo* type strain, PXO99A, and 99.65% with the *Xoo* traditional representative strain, YC11, in the Yangtze River region (Table 2). The chromosome contains 3630 protein coding genes, 53 transfer RNAs (tRNAs), 6 ribosomal RNAs (rRNAs), 144 non-coding RNAs (ncRNAs), 811 pseudogenes and 14 *Tal* genes (Table 2). The circular representation of the complete genome of LA20 is shown in Appendix A.

### 2.3. Tal Genes of Xoo Strain LA20

Despite having a sequence similarity of up to 99.65% with the genome sequences of LA20 and YC11 (Table 2), the data show that there are 10,900 single nucleotide polymorphisms, 188 deletions and 199 insertions (Appendix A). These genomic variants may result in the more 16 protein coding genes, 89 pseudogenes and 2 *Tal* genes in the genome of LA20 (Table 2).

The type and number of *Tal* genes are very critical to *Xoo* pathogenicity and rice resistance. It was found that the *Tal* genes between LA20 and YC11 are highly diverse (Figure 2). In LA20, 14 *Tal* genes was identified, which has 2 more than the number of those from YC11; *Tal1b*, *Tal2b*, *Tal3c*, *Tal3b*, *Tal3a* and *Tal4* were found in LA20 and absent in YC11 (Table 3). Among the above six *Tal* genes, *Tal1b* (*pthXo1*) and *Tal4* (*pthXo6*) have been known to target *OsSWEET11* (*Xa13*) and bZIP transcription factor-encoding gene (*OsTFX1*), inducing rice susceptibility, respectively (Table 3). *Tal2b*- and *Tal3b*-encoding interfering TALEs (iTALEs) are brand new, showing no homology with the reported TALEs (Table 3). *Tal3c* and *Tal3a* in LA20 are identical to *Tal5b* and *Tal5a* in PXO99A (Table 3). Further, *pthXo3* (*avrXa7*), which targets *Xa7* inducing rice resistance, was only found in YC11, but not in LA20 (Table 3). Therefore, we speculated that the diversity of TALEs type in LA20, such as special ones, *Tal1b* (*pthXo1*)/*Tal4* (*pthXo6*), novel ones, *Tal2b* and *Tal3b* (iTALEs) and the lack of *pthXo3* (*avrXa7*), may cause re-epidemics of BB, replacing the traditional representative strain, YC11, in the Yangtze River region, China.

## 3. Discussion

Recently, BB re-emerged, and there was a break out in the Yangtze River region, which is the major rice-producing region in China. In this study, the representative *Xoo* strain, LA20, was isolated and identified from re-epidemic BB samples. It was demonstrated that LA20 is a Chinese R9 *Xoo* strain that can override *xa5*- and *xa13*-mediated resistance. Moreover, LA20 can also overcome *Xa7*-mediated resistance. The complete genome further revealed variability in *Tal* genes in the pandemic *Xoo* strain, LA20, compared with that of the traditional representative strain, YC11 (R5).

The traditional dominant *Xoo* strain, YC11 (R5), was replaced by the newly emerging one, LA20 (R9), in the Yangtze River region. It is believed that during the long-term co-evolution between pathogen and host, the arms race might be disequilibrated through either the evolution of virulence factors of pathogens or the loss of host resistance genes. TALEs are the most important virulence factors of *Xoo*. An individual *Xoo* strain usually contains 9–21 *Tal* genes [20]. There are two more *Tal* genes in LA20 than there are in YC11. Further, *Tal1b*, *Tal2b*, *Tal3c*, *Tal3b*, *Tal3a* and *Tal4* were found in LA20 and absent in YC11. These variations from TALEs may have contributed to the BB re-outbreak.

BB *R* gene, *xa5*, *xa13* and *Xa7* are usually employed to breed BB-resistant varieties in the Yangtze River region, China. The former YC11 (R5) from the Yangtze River region cannot overcome *xa5*- and *xa13*-mediated BB resistance. However, the special *Tal1b* (*pthXo1*) and *Tal4* (*pthXo6*) of LA20 can recognize and bind to the promoter of the dominant *Xa5* and *Xa13*, overriding *xa5*- and *xa13*-mediated BB resistance [21,22,23,24]. Furthermore, *pthXo3* in YC11 can bind to the promoter of *Xa7* and induce its expression to trigger the defense [25]. Conversely, *pthXo3* was not found in LA20, and LA20 broke down the *Xa7*-mediated resistance. Further, it is interesting that two new iTALEs was identified in LA20. iTALEs contain 45 or 129 bp deletions in the sequence encoding the N-terminal region and lack the C-terminal AD domains, unlike typical TALEs [26,27]. It was reported that iTALEs suppressed *Xa1* resistance triggered by typical TALEs [28]. These variations may result in LA20 escaping immunity mediated by *xa5*, *xa13, Xa7* and *Xa1.* Analyzing virulence factors will facilitate the further understanding of the co-evolution of rice and *Xoo*, as well as the developing of strategies to clone new disease resistance genes for rice breeding.

In this study, the combination of PacBio and Illumina sequencing quickly, effectively, and accurately conferred the complete gap-free genome of LA20. HiFi long-read sequences with read lengths averaging 10–25 kb and accuracies >99.5% revealed the instability and variability in containing nearly identical tandem-repeat domain *Tal* genes in the epidemic variant, LA20. The approach and genome data will contribute to assessing the epidemic disease status and the valuable resource for a better understanding of pathogen evolution and the molecular pathogenesis of *Xoo* that caused the BB resurgence in the Yangtze River region, China.

## 4. Materials and Methods

### 4.1. Isolation and Identification of Xoo

Infected leaves of the rice variety “Quanyou1606” were successively sterilized with 70% ethanol and 1% sodium hypochlorite for 30–60 s, rinsed with distilled water for three times, then cut, crushed and soaked in 1 mL distilled water for 10 min. The suspension was streaked and cultured on peptone sugar agar (PSA) medium for 72 h at 28 °C. Circular, smooth-margined, convex colonies were selected for further purification and molecular identification.

For molecular identification, *Xoo*-specific polymerase chain reaction (PCR) was carried out using *16S rRNA* gene primers 16SrRNAF (5′-AGAGTTTGATCATGGCTCAG-3′)/16SrRNAR (5′-AAGGAGGTGATCCAGCCGC-3′) with target fragment 1539 bp, and *HP* gene primers Xoo163F (5′-CAATGCACACGTGGAAAGGG-3′)/Xoo163R (5′-CTTGCAAGGGATAGAAGCGT-3′) with target fragment 163 bp. Finally, amplified fragments were sequenced, and we performed sequence identity analysis using NCBI Blastn.

### 4.2. Rice Materials

The differential rice varieties used in this study were a simplified combination of IRBB2, IRBB3, IRBB5, IRBB13, IRBB14 and IR24, which were provided by the International Rice Research Institute (IRRI), Philippines, via National Rice Germplasm Genebank, China. All rice varieties were planted in the experimental field in China National Rice Research Institute. Plants in the booting stage were used for artificial *Xoo* inoculation assays.

### 4.3. Pathotype Analysis

The booting stage near-isogenic lines harboring individual *Xa*-resistance gene were inoculated with LA20 (OD_600_ = 0.5) by a leaf clipping method [29]. The disease was scored as the percent lesion area (lesion length/leaf length) at 21 days after inoculation, as in our previous study. The ratios of lesion length to entire leaf length of <1/4 and ≥1/4 were classified as resistant (R) and susceptible (S), respectively [30]. All experiments were repeated three times. The least significant difference (LSD method) was used for data analysis. A pathotype classification of LA20 was performed according to their reactions on NIL rice (Appendix A).

### 4.4. Genome Sequencing, Assembly and Annotation

Genomic DNA of LA20 was extracted using SDS method [31]. The SMRT bell TM Template kit (version 2.0) and NEBNext^®^Ultra™ DNA Library Prep Kit for Illumina ( New England Biolabs Inc., Beverly, MA, USA) were used to construct sequencing library for PacBio Seque1and Illumina NovaSeq PE150 sequencing by Novogene (Beijing, China) [32]. The average fragment length of the sequencing library is about 10 kb for circular consensus sequencing (CCS) to achieve long and high fidelity (HiFi) reads. HiFi reads were de novo assembled with Canu (v2.0) and Racon (v1.4.13) [33]. Reads from Illumina were used to correct genome sequences for improving the quality of assembly using Pilon software (v1.22) [34]. Circos software was used to cyclize and adjust the starting site [35]. Finally, the assembly was annotated using the NCBI Prokaryotic Genome Annotation Pipeline [36] and GeneMarkS (v4.17) [37].

### 4.5. Comparative Genomics and TALEs Analysis

For structural comparison, complete genomes were aligned using progressive Mauve [38] with default settings. For TALEs analysis, *Tal* genes and RVD sequences were assessed using AnnoTALE tools according to a previous report [19].

## 5. Conclusions

We demonstrated that the causative agent of newly emerging BB in the Yangtze River region is the Chinese R9 *Xoo* strain, which is able to override the widely adopted *xa5-*, *Xa7-* and *xa13*-mediated resistance in rice varieties in the Yangtze River region. Further, we provide an HiFi long-read gap-free genome sequence of LA20 and identified 14 *Tal* genes. The variability in TALs in their numbers and sequences compared with those of the former representative strain, YC11 (R5), in the Yangtze River region may have caused the re-outbreak of BB.

## Figures and Tables

**Figure 1 ijms-24-08132-f001:**
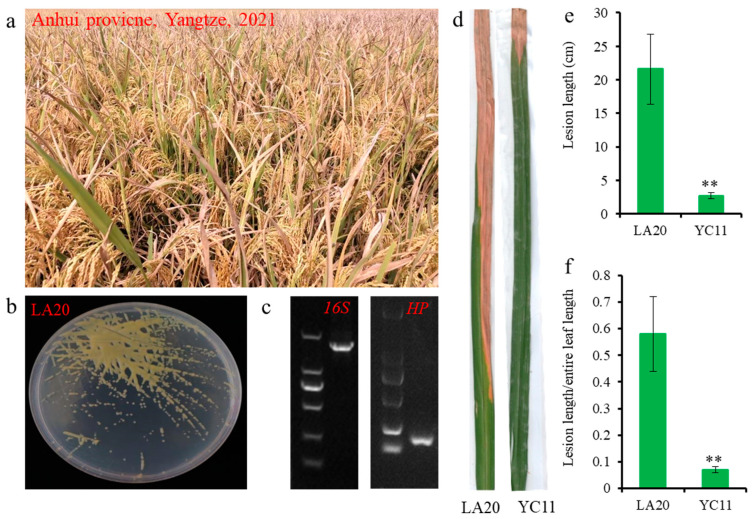
Identification and pathotype analysis of the newly emerging *Xoo* strain, LA20. (**a**) The field symptoms of BB in Anhui province, Yangtze, in 2021; rice variety: Quanyou1606. (**b**) colony morphology of LA20. (**c**) PCR amplification of *16SrRNA* and *HP* gene of LA20. (**d**–**f**) resistance evaluation of IRBB7 to *Xoo* strains LA20 and YC11. (**d**): leaves’ phenotype. (**e**,**f**): lesion length and ratio of lesion length to entire leaf length of IRBB7 inoculated with LA20 and YC11 at 21st day, respectively. Error bar: SD from 18 inoculated leaves. Significance of differences was determined by Student’s *t* test, ** *p* < 0.01.

**Figure 2 ijms-24-08132-f002:**
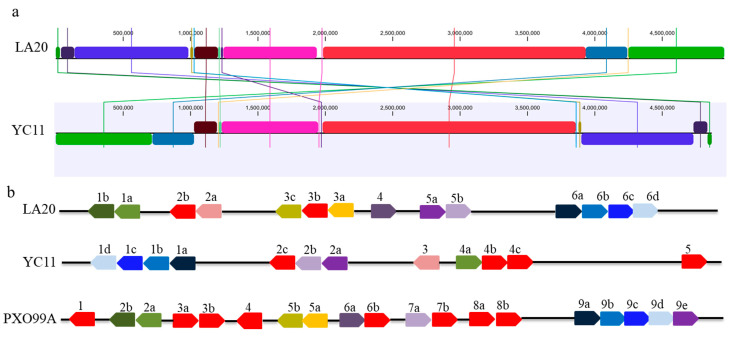
Comparison of whole genome and TALEs in *Xoo* strains. (**a**): Progressive mauve alignment chromosome of LA20 and YC11. The ruler indicates distance from the annotated origin in base pairs. (**b**): The *Tal* genes of *Xoo* strains. The genes are represented as arrows at their relative position in the linearized chromosome. Arabic numbers indicate the serial number of *Tal* gene clusters in a *Xoo* strain. Lowercase letter indicates each *Tal* gene in a certain gene cluster. The identical effectors with identical RVD sequences are lined with the same colors. The strain-specific effectors are indicated in red for LA20, YC11 and PXO99A.

**Table 1 ijms-24-08132-t001:** Pathotype of *Xoo* strain, LA20, in six differential rice varieties harboring individual resistance gene.

Differential Variety	Resistance Gene	Mean Lesion Length (cm)	Ratio of Lesion Length to Entire Leaf Length	Resistance Evaluation
**IRBB2**	*Xa2*	23.3 ± 3.64	0.61 ± 0.10	S
**IRBB3**	*Xa3*	25.2 ± 5.47	0.65 ± 0.14	S
**IRBB5**	*xa5*	16.1 ± 4.60	0.46 ± 0.13	S
**IRBB13**	*xa13*	15.4 ± 2.43	0.41 ± 0.06	S
**IRBB14**	*Xa14*	20.9 ± 2.47	0.56 ± 0.13	S
**IR24**	*Xa18*	27.0 ± 4.61	0.73 ± 0. 13	S

S: susceptible (ratio from 0.25 to 1) were measured 21 days after inoculation. ±: SD from 18 inoculated leaves.

**Table 2 ijms-24-08132-t002:** Genome comparison between *Xoo* strains, LA20 and YC11.

Genome Features	LA20 (R9)	YC11 (R5)
**Accession No.**	CP114600	CP031464
**Size (bp)**	4,960,087	4,867,200
**GC content (%)**	63.69	63.74
**Protein coding genes**	3630	3614
**tRNAs**	53	53
**rRNAs**	6	6
**ncRNAs**	144	146
**Pseudogenes**	811	722
***Tal* genes**	14	12
**ANI (%)**	99.67	99.65

**Table 3 ijms-24-08132-t003:** *Tal* genes and RVD sequences in LA20 and similar genes in YC11 and PXO99A.

Genes	RVD Sequences	Function	Targeted Genes	YC11	PXO99A
** *Tal1b* **	NN-HD-NI-NG-HD-NG-N *-HD-HD-NI-NG-NG-NI-HD-NG-NN-NG-NI-NI-NI-NI-N*-NS-N*	*pthXo1*	*OsSWEET11(Xa13)*	*/*	*Tal2b*
** *Tal1a* **	NI-NG-NN-NG-NK-NG-NI-NN-NI-NN-NI-NN-NS-NG-NS-*NN*-*NI*-*N**-*NS*-*NG*	/	*/*	*Tal4a*	*Tal2a*
** *Tal2b* **	NS-NG-NG-NG-NG-HD-H*	*iTal*	/	/	/
** *Tal2a* **	NS-HD-NG-NG-HG-NG-HD-HD-NG-HD-NN-HD-NG-HD-NI-NI-NI-N*	*iTal*	*/*	*Tal3*	/
** *Tal3c* **	NI-H*-NI-NN-NN-NN-NN-NN-HD-NI-HD-HG-HD-NI-N*-NS-NI-NI-HG-HD-NS-NS-NG	/	*/*	*/*	*Tal5b*
** *Tal3b* **	NS-HG-HG-HD-NS-NG-HD-NN-NG-HG-NG-HD-HG-HD-HD-NI-NN-NG	*iTal*	/	/	/
** *Tal3a* **	NI-NS-HD-HG-NS-NN-HD-H*-NG-NN-NN-HD-HD-NG-HD-NG	/	*/*	*/*	*Tal5a*
** *Tal4* **	NI-N*-NI-NS-NN-NG-NN-NS-N*-NS-NN-NS-N*-NI-HG-HD-NI-HD-HD-NG	*pthXo6*	*OsTFX1*	*/*	*Tal6a*
** *Tal5a* **	NN-HD-NS-NG-HD-NN-N*-NI-HD-NS-HD-NN-HD-NN-HD-NN-NN-NN-NN-NN-NN-NN-HD-NG	/	*/*	*Tal2a*	*Tal9e*
** *Tal5b* **	NI-HG-NI-NI-NI-NN-HD-NS-NN-NS-NN-HD-NN-NI-HD-NN-NI-NG-*HD*-*NG*	/	*/*	*Tal2b*	*Tal7a*
** *Tal6a* **	HD-HD-HD-NG-N*-NN-HD-HD-N*-NI-NI-NN-HD-HI-ND-HD-NI-HD-NG-NG	*pthXo8*	*OsHEN1*	*Tal1a*	*Tal9a*
** *Tal6b* **	HD-HD-NN-NN-NI-NG-HD-S*-HG-HD-NG-N*-NG-HD-HD-N*-NI-NI-NN-HD-HI-ND-HD-NG-NN-HG-N*	*avrXa23*	*Xa23*	*Tal1b*	*Tal9b*
** *Tal6c* **	NI-NN-N*-NG-NS-NN-NN-NN-NI-NN-NI-NG-HD-HD-NI-HG-N*	*avrXa27*	*Xa27*	*Tal1c*	*Tal9c*
** *Tal6d* **	NI-NN-NI-HG-HG-HD-NG-HD-HG-HD-HD-HD-NG	/	*/*	*Tal1d*	*Tal9d*
** */* **	NI-HG-NI-HG-NI-NI-NI-HD-NN-HD-HD-HD-NG-HD-N*-NI-HD-HD-NN-NS-NI-NN-NN-NG-NN-HD-N*-NS-N*	*pthXo3*	*Xa7*	*Tal4c*	/

* predicted deletion of the thirteen codon in the repeats; / absent. *Tal* genes and RVD sequences are based on AnnoTALE [19].

## Data Availability

The whole-genome sequences reported here have been deposited in GenBank (https://www.ncbi.nlm.nih.gov/, accessed on 23 December 2022) under the accession number CP114600 (BioProject: PRJNA913657, BioSample: SAMN32303221).

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
