# Peer review of "Complete Genomic Sequence of Xanthomonas oryzae pv. oryzae Strain, LA20, for Studying Resurgence of Rice Bacterial Blight in the Yangtze River Region, China"

_ijms, 2023, doi:10.3390/ijms24098132_

Round 1

Reviewer 1 Report

Identification of new variants of Xanthomonas oryzae pv. oryzae (Xoo), the causative agent of bacterial blight in rice, is important to maintain rice plant population productivity. The authors present the results of genome-wide analysis of the newly emerging Xoo strain LA20. The authors found variations in the TALE family of transcriptional activation genes to overcome mechanisms of rice resistance to bacterial blight. In general, the results are not questionable. The authors used data from two sequencing platforms from Illumina and PacBio to assemble the genome de novo and obtain data on the variability of containing nearly identical tandem-repeat domain of the Tal genes. The manuscript is written compactly and concisely. The results are illustrated in three figures and three tables, although the resolution of the figures could have been higher.

I have a few comments on the design of the article.

First, the manuscript must contain a direct reference to the Xoo strain LA20 genome sequence. In the Data availability section, the authors indicate the Bioproject number, but his check on the link

https://www.ncbi.nlm.nih.gov/bioproject/?term=PRJNA224116

leads to a general Refseq Prokaryotic Genome Annotation Project.

Similarly, checking accession number CP114600 https://www.ncbi.nlm.nih.gov/genome/?term=CP114600

does not result in a reference to the genome sequenced in this study

Abstract, lines 16-17.

Please provide more results on the variations found in the Tal genes in the LA20 strain and how they affect overcoming xa5, xa13 and Xa7 mediated resistance. This is the most significant result of this study.

Page 4. Figure 2.

I'm not sure that this figure is significant information, it's more of an illustration. Perhaps it would be more appropriate to include it in the supplementary material. What is written in the center of the figure? 'The totle length???'

Lines 86-89. This information is more appropriate for Section 4.4 of the 'Materials and Methods'.

Finally, the manuscript contains many references, e.g., Chen et al., 2019, Tatusova et al. 2016, Besemer et al. 2001, etc., which are not listed in the reference list. This part needs substantial refinement.

Please note the rigor of the key terms of the article. For example, the abstract specifies TALs, while most of the article specifies TALEs or Tal genes.

Reviewer 2 Report

In the current manuscript, Yuxuan Hou and colleagues present a complete genomic sequence of Xanthomonas oryzae pv. oryzae Strain LA20. Overall, the manuscript is quite minimalistic, having only 200 rows of main text in total, but this could be ok, considering the Communication type of submission. The manuscript is well and logically structured.

Regarding scientific content, I believe that the high-quality assembly authors provided (using a combination of Illumina and PacBio reads) is useful for the community of Rice researchers. The resulting single genomic circular DNA of length 4,960,087 bp is appropriately deposited in the NCBI nucleotide database.

However, some issues should be solved prior to publication:

1.) R (resistance) and RR (repeat region) abbreviations could be quite confusing for the readers considering both express different things

2.) Line 39 ... racial discrimination ... another term should be used, variety discrimination maybe (?)

3.) Figure 1e and 1f ... I would recommend using some statistical test to prove observed differences are statistically significant (e.g. Wilcoxon test) and also present p-values and asterisks

4.) Line 83, please check this "(). ±: SD" 

5.) Discussion could be a little more complex

line 93 ... Chromsome

line 130 ...The Genes should be The genes

Round 2

Reviewer 2 Report

Dear Authors, thank you for the revised version of your manuscripts. You have successfully fixed all my issues, maybe just one point, please unify the References, i.e. journal names should be always abbreviated and doi numbers should be included as well.
